# External childcare and socio-behavioral development in Switzerland: Long-term relations from childhood into young adulthood

**Margit Averdijk**[1]*, **Denis Ribeaud**[1], **Manuel Eisner**[1,2]

**1** Jacobs Center for Productive Youth Development, University of Zurich, Zurich, Switzerland, **2** Institute of Criminology/Violence Research Centre, University of Cambridge, Cambridge, United Kingdom

* margit.averdijk@jacobscenter.uzh.ch

## Abstract

This study examined early external childcare in relation to development from age 7 to 20. A Swiss sample was used ($N$ = 1,225; 52% male). Development included multi-informant-reported externalizing behavior, internalizing problems, prosocial behavior, delinquency, and substance use. Growth curve models revealed that, dependent on the informant, time in a daycare center was related to increased externalizing and internalizing problems until at least age 11. It was not related to delinquency. Roughly three days per week at a daycare mother or playgroup was related to increased externalizing behavior. External family care was associated with increased prosocial behavior. Finally, time in a daycare center was associated with fewer externalizing but more internalizing problems and substance use for children from vulnerable backgrounds. This relation with substance use lasted to age 20.

## Introduction

According to the notion of developmental pathways, early life experiences can shape lives for a long time to come, affecting not only initial development but also long-term socio-behavioral trajectories [1]. The contextual embeddedness of experiences plays a crucial role in the ultimate stability and change of behavior [2]. Over the past decades, one context that has increasingly contributed to shaping lives is external childcare. For many children in Western countries, attending external childcare is common. The increasing demand for and supply of external childcare is a topic that has received much attention in the political debate, media, and public opinion, because it not only touches on family policies and early education but is also intimately linked to the ability of mothers to work outside of the home.

One often asked and studied question is how external childcare is related to child development. An increasing number of studies has examined this question but, though contributing greatly to our knowledge, most offer only a momentary glimpse of development because the underlying data are limited to snapshots of pathways. Specifically, the vast majority of studies could only examine short-term relations of external childcare with development. Few were

**Funding:** The z-proso study on which the research reported in this manuscript is based was financially supported by the Swiss National Science Foundation (Grants 405240-69025, 100013_116829, 100014_132124, 100014_149979, 100014_149979, 10FI14_170409/1, 10FI14_170409/2, 10FI14_198052/1), the Jacobs Foundation (Grants 2010-888, 2013-1081-1), the Jacobs Center for Productive Youth Development, the Swiss Federal Office of Public Health (Grants 2.001391, 8.000665), the Canton of Zurich's Department of Education, the Swiss Federal Commission on Migration (Grants 03-901 (IMES), E-05-1076), the Julius Baer Foundation, and the Visana Foundation. The research reported in this manuscript was supported by a grant from the Jacobs Foundation, awarded to MA. The funders had no role in the study design, data collection and analysis, decision to publish, or preparation of the manuscript.

**Competing interests:** The authors have declared that no competing interests exist.

able to assess long-term relations, because the necessary data are often not available. Thus, little is known about the long-term relation of external childcare with behavior.

Two other aspects that past studies have often overlooked relate to the multi-faceted nature of both external childcare and socio-behavioral development. For example, many, though not all, studies have focused primarily on daycare centers, though many children attend other types of (sometimes informal) external care such as by external family members. In addition, because many studies on external childcare have focused on externalizing behavior, other behaviors remain understudied. These include sub-types of externalizing behavior, including aggressive and non-aggressive, but also prosocial behavior, internalizing problems, and manifestations of problem behavior in adolescence and beyond, notably delinquency and substance use.

In this paper, we addressed these issues by using longitudinal data from Switzerland that capture behavioral development from school entry to early adulthood. Our data include a range of external childcare arrangements as well as behavioral aspects, and enable us to examine the relation of external childcare before Kindergarten with child and youth development from age 7 to 20. We are not aware of other studies that included a similarly wide range of socio-behavioral outcomes, types of childcare, and, particularly, length of follow-up. In order to account for the possibility that our results are spurious and explained by third factors, we controlled for various confounding factors. Nevertheless, like other studies on the same topic, our study is observational and we therefore cannot claim that the remaining relations are more than correlational.

We focused on the amount and type of external childcare but not on its quality, which is another aspect that has been related to child development but was not included in our data [3]. This is a limitation of our study, as previous studies have found some support for the notion that children visiting very high quality centers showed less externalizing behavior than children who visited low- or regular-quality centers [4, 5]. On the other hand, prior research has also shown that the relation between childcare quantity and behavior cannot be accounted for by the quality of care as more time in external care was related to more externalizing behavior at all levels of quality [6].

## Relations between the quantity of external childcare and social development

Past literature reviews on the link between the quantity of external childcare and child development have concluded that external childcare is related to increased problem behavior. For example, Belsky [7] famously argued that "early, extensive, and continuous nonmaternal care" is associated with developmental risks, whereas Huston et al. [8] concluded that the evidence linking early, extensive nonparental care and teacher-reported externalizing behavior is persuasive. However, despite an increasing amount of studies, literature reviews also recognize that there are inconsistencies between studies and that much is still unknown (e.g., [8, 9]). For example, it is currently unclear to what extent study results are causal or affected by selection bias [10], whether conclusions extend to all or only some forms of external childcare and to different types of problem behavior, whether the link between external childcare and problem behavior represents the populations most at risk, and how the relation develops across the life course.

Regarding the type of childcare, most prior research in the field has focused on a particular type of external childcare, namely on care in daycare centers, whereas not many studies have explicitly examined the relation between other types of external childcare and problem behavior (for exceptions, see [11–14]. Although some found no [15, 16] or even a negative relation

[17] between the two, the majority linked daycare center attendance to more problem behavior. For example, studies from the U.S. [6, 13, 18, 19], Austria [20], the Netherlands (e.g., [21]), and Australia [22] associated center care with lower self-control, weaker interpersonal skills, and more externalizing behavior. With one exception [13], all of these studies used regression-based analyses with covariate adjustment. Some studies have suggested that the effects of non-parental care may take some time to manifest [23] as findings from studies on the relation between the amount of external childcare and externalizing or prosocial behavior at ages 1.5 to 3 years has been conflicting, with some finding no relation [24–26] and others that have [5, 27, 28]. While some of these studies have used standard regression designs, others [5, 26, 28] have gone beyond these and used more robust methods.

Several potential explanations have been offered for the found relations. Though early discussions questioned whether external childcare reduced safe child-parent attachment [29], it has more broadly been related to less positive mother-child interactions (e.g., [30]). In addition, both the experienced separation of the child from its parents and the presence of other children in center-based childcare may increase children's level of stress. Indeed, prior research has shown that children's cortisol levels, a marker for stress, are increased in external childcare, especially daycare centers (e.g., [31]). Furthermore, children in center-based care are surrounded by other children, and although peer interactions can be beneficial, problem behavior in center-based settings can be "contagious," and children can learn problem behavior from each other [32]. Also, displaying problem behavior can be a way to get attention from caregivers.

The existing literature has focused more on externalizing behavior than on internalizing problems. Studies that have examined the link between external childcare and internalizing problems using various methods (including both traditional regression-based methods and more robust quasi-experimental techniques) have yielded mixed results. Some studies found some evidence that more time in a daycare center was related to more internalizing problems [28, 33], whereas others found no relation (e.g., [21]) or even that a larger amount of time spent in center-based care was associated with fewer internalizing problems [22, 34]. Thus, conclusions in this area are still open.

## Validity and selection bias

Another question that is still open is to what extent the relation between external childcare and behavior is causal. Particularly, researchers in the field have argued that "too little attention to internal and external validity may be biasing much of the scientific and public thinking on the topic" [10, p. 133]. One difficulty with prior studies on external childcare is that they are not experimental and therefore affected by selection bias: Whether or not children attend external childcare is often associated with certain characteristics of the children and their families, which in turn may explain away the relation between external childcare and problem behavior. Most studies (including ours) have attempted to control for omitted variable bias by controlling for a range of background characteristics, such as a family's socio-economic status and parenting characteristics but potential bias from unmeasured characteristics remains. Other studies have gone beyond ordinary regressions using covariate adjustment, such as sibling comparisons, propensity score matching, and instrumental variables. Although each method has their own unique advantages and disadvantages and cannot match a causal experiment, these methods yield less biased results.

The studies that have used these methods have produced inconsistent results as well. In their review of these studies, Dearing and Zachrisson [10] concluded that in six of the studies, time in childcare predicted more problems, that one predicted the opposite, and that six found

null results. For example, a US-based study found that early non-maternal care initiated in the first three years of life was associated with less problem behavior in childhood and early adolescence when comparing families [35]. However, when comparing siblings within families, they found no differences between siblings with different external care experiences or that those who had entered non-maternal care had more behavior problems. Another US-based study among low-income families with pre-school children found that using instrumental variables techniques in some cases flipped the relation between externalizing behavior and center care so that center care was associated with less externalizing problems [36]. However, two other US-based studies found that external care was related to increased behavior problems among children regardless of whether the researchers used traditional regression analyses with covariate adjustment or propensity score matching, instrumental variables, or fixed effects methods [13, 37].

Another way in which studies have tried to circumvent selection bias is by analyzing sudden policy changes that affect a certain part of a country but not another. An example is a study in Quebec, Canada, where researchers have examined the consequences of a policy change that involved the provision of a universal subsidized child care program. It was found that the increased use of external childcare due to this policy change was associated with more anxiety, more physical aggression and opposition, and lower standardized motor and social development in early childhood, although there were no associations with hyperactivity-inattention or separation anxiety [33].

It is difficult to arrive at an overall conclusion regarding the causality of the link, which is evidenced by the fact that in one prior review, the authors were not very confident that the relation between childcare quantity and problem behavior is causal [10], whereas others concluded that with a few exceptions, the causality of the relation between external care and problem behavior is supported and robust to methodological variety [8, p. 628]. Summing up, prior studies on the short-term relation between external childcare and behavior have been mixed and warrant careful interpretation of conclusions. One question that has arisen is whether moderators, most notably children's backgrounds, might explain the differences in results.

## Risk, childcare, and social behavior

The way in which early experiences shape trajectories is not just dependent on the experience itself and the way it affects subsequent events but also a function of the larger environment in which children develop [1]. For example, reviews have recognized that external childcare must "be viewed in the context of children's intrinsic characteristics, developmental trajectories, and other experiences" [38, p. 999]. Specifically, the relation between external childcare and problem behavior may vary by familial background to the extent that families with high levels of stress and/or where parenting strategies to reduce problem behavior are inadequate, the structure and rule-setting of an external childcare environment may help children to improve their behavior. Indeed, there has been some, though conflicting (e.g., [28]), evidence for this: Although some studies have found no evidence that the relation between external childcare and behavior differs between children from vulnerable families (with e.g., negative parenting strategies and/or low income) and other children [5, 13, 39–41], other studies have found differential relations: For children from vulnerable backgrounds, attending daycare may reduce problem behavior, whereas children from average or nonvulnerable backgrounds may show no difference or more problem behavior [25, 34, 42–44]. For example, two studies from Canada found that children from low-risk families had lower physical aggression scores at age 4 when they had received maternal care than when they had received nonmaternal care in infancy; on the other hand, children from the 16% of families that had the highest risk had

higher physical aggression in home care than in external daycare [45, 46]. There are gaps, though, in our understanding of how these findings play out as children get older.

## Relations over the life course

The question of how long relations (in either direction) continue to exist over the life course remains largely unanswered. Although it can generally be expected that more recent life experiences play a larger role than more distant ones and that the relation between early external childcare and development decreases as children get older, life course perspectives also suggest that early differences can exacerbate later in life. For example, the notion of developmental cascades posits that children's exposure to early risks may contribute to other risks with age [47]. On the other hand, early skills and abilities can compile and compound over the life course and therefore condition children's later development [48]. Early educational interventions for children from vulnerable backgrounds to improve skills and later life opportunities (e.g., Head Start, the Perry Preschool Project) have been stimulated based on this idea [49], but few studies have examined whether more typical early childcare is related to behavioral trajectories into the teenage years and beyond, despite its potential life course significance. Those that have suggest that the relation may persist until the end of primary school [20], although another linked external childcare to less externalizing behavior in adolescence [44]. Longer follow-ups into early adulthood are rare and have produced mixed results. One exception is the long-term follow-up in Quebec by Baker et al. [50], who, as a natural experiment, analyzed the consequences of the earlier mentioned policy change that involved universal subsidized childcare. Results showed that the relation between increased childcare use and children's social behavior persisted into the school years (ages 5 to 9) and in some cases became larger. The policy change was associated with more hyperactivity, anxiety, aggression, indirect aggression, and a worse relationship with a teacher. At ages 12 to 20, external childcare was associated with worse self-reported health and overall life satisfaction, but not with mental health. Furthermore, exposure to the Quebec program was associated with around 20% higher crime rates at ages 12 to 20. Another exception is the follow-up of the previously mentioned NICHD study at age 18 [51]. In contrast to reports at prior ages, the researchers reported that center care was related to less risk taking and greater impulse control among females at age 18, underscoring the importance of examining the potential long-term relations between external childcare and development.

## External childcare in Switzerland

In the current study, our goal was to examine the long-term relation of various forms of external childcare with social behavior. We examined a range of externalizing behaviors, prosocial behavior, internalizing problems, delinquency, and substance use.

Researchers have argued that international replication has been underutilized as a scientific tool to examine the link between external childcare and behavior and that it is important to grow the international research base so that ultimately, individual dimensions of economic and sociopolitical contexts can be isolated [10]. Our study helped growing this international research base by using data from a Swiss longitudinal study, covering ages 7 to 20. Although female labor participation in Switzerland is high (79% of mothers of under 12-year-olds have a paid job), most mothers of under 12-year-olds (81%) work part-time [52] and many children attend childcare on a part-time basis [53]. Among fathers of under 12-year-olds, only a minority (12%) works part-time. According to the most recent overview of the Swiss Federal Statistical Office, 72% of 0- to 3-year-olds in Switzerland attend some type of external childcare. In total, 41% attend institutional childcare and 51% non-institutional childcare [54]. Among those children who attend institutional childcare, most do this for 10 to 29 hours per week.

Compared to other European countries (though less so to other countries), Switzerland has been rated relatively poorly on indicators of national family-friendly policies (e.g., paid leave for mothers and fathers), ranking in the bottom three along with Cyprus and Greece [55]. It scored relatively high, on the other hand, on rates of breastfeeding, and average on childcare enrolment under 3. Although external childcare is expensive, only 5% of parents with under-3-year-olds cite affordability as the main reason for not making (more) use of it [55]. A Swiss study among a representative sample of 1,000 Swiss participants performed a so-called "choice experiment" where participants could choose between different forms and quantities of external childcare. Results based on regression models showed that an increased availability of external childcare provisions would affect parental labor participation only marginally [56].

Little is known about the relation between external childcare and development in Switzerland. To our knowledge, four other studies exist. Three found that center-based care was related to more externalizing but not internalizing problems [57–59]. A fourth study focused on motor, cognitive, language, and social skills but did not measure problem behavior. Toddlers who attended a daycare center for at least two days a week did not differ from children cared for by family members on the measured domains [60]. All four studies (based on regression models) included only few control variables and a sample of relatively advantaged families. Our study was based on a large sample of children in the largest city in Switzerland. In a prior publication, it was concluded that, after controlling for a range of covariates, the quantity of external childcare was related to increased externalizing and internalizing problems at age 7 [61]. This publication formed the starting point for the current analysis that includes follow-ups until age 20.

## Method

### Sample

Data were drawn from the Zurich Project on the Social Development from Childhood into Adulthood (z-proso), a combined longitudinal and intervention study [62]. The target population consisted of all 2,520 children who entered the first grade of public primary school in a large Swiss city in August 2004. Sampling was based on a cluster-randomized approach with schools as the randomization units. All 90 public primary schools in the city were classified by school size and socio-economic background. A stratified sample of 56 schools was drawn. The target sample consisted of all 1,675 first graders. 94% of the target sample participated in at least one of the data collections. Participation and retention rates per data collection and informant are displayed in S1 Table. In 46% of all cases, both parents were born outside of Switzerland. This high rate of immigrants in the sample is representative of the city. Our analyses did not include control variables for the interventions, because these took place after the period of external childcare and had minimal effects [63, 64].

We used data from the youths, their parents, and their teachers at ages 7 through 20. Data collection followed local data protection regulations. Active parental consent was obtained before the data collection in grade 1 and again before the data collection in grade 5. The youths gave their active informed consent starting grade 7. The parents could opt their children out of the data collections in grades 7 and 9.

The parents received shopping vouchers the equivalent of US$30 for their participation. At ages 13, 15, 17, and 20 the youths received an incentive worth the equivalent of US$30, US$50, US$60, and US$75 at each respective wave in exchange for their participation. The size of the incentives was based on wage levels for Swiss youths. For the teachers, participation in the first three years was compulsory. Starting in grade 5, they received a book voucher worth the equivalent of US$50 when they completed more than six questionnaires.

Computer-assisted parent interviews, ceased after age 11, took place in the respondent's home. Since the mother tongue of around half of the parents was not German, contact letters, telephone recruitment, informed consent, and interviews were made available in nine additional languages. Teachers completed a questionnaire and returned it by mail. Since the frequency of interactions between teachers and the youths significantly reduced after age 15, we analyzed teacher data until that age. For the children, the first three data collections were done in the form of computer-assisted personal interviews conducted by trained interviewers at school (approx. 45 minutes). Starting in grade 5, the youths completed a written questionnaire of approximately 90 minutes in a school setting. At age 20, they completed a computer-assisted self-interview in a lab setting.

## External childcare

We defined external childcare as care outside of the child's home. Thus, care by live-in nannies was excluded. Similar to prior studies (e.g., [28, 39, 65, 66]), data on childcare were obtained retrospectively. The reliability of such reports is supported [66]. Maximizing reliability and validity, an Event History Calendar (EHC) was used at age 7 to collect time-ordered data about the first years of life as EHCs encourage sequencing and parallel retrieval of events (e.g., [67]). In 94% of cases, the mothers completed the interview. More information about the EHC is found in Eisner et al. [68]. The EHC was designed as a grid with thematic domains (such as household composition, childcare arrangements, etc.) represented by the rows, and time periods (three months) represented by the columns. It was organized top-down, with the most easily recalled themes mentioned first, followed by themes that were more difficult to remember. By addressing the events and their interrelated nature, respondents are more likely to remember isolated events and resolve inconsistencies across the domains.

We acknowledge the possibility that retrospective self-reported data can introduce bias. Two papers have examined the validity of the current EHC. The first examined its criterion-related validity. It was concluded that risk factors measured on the EHC were correlated with behavior outcomes in the expected direction, that the size-order and relative importance of early risk factors were in line with the previous literature, that longer exposure was associated with an added risk, and that the likelihood of problematic outcomes was related to cumulative contextual risk [68]. The second examined the EHC data on household composition and found 92% to 96% agreement between the EHC and the regular Wave 1 data [69].

For all quarters prior to completion of the EHC, respondents estimated how many days per week a child received external childcare by external family members, acquaintances or neighbors, daycare mothers, daycare centers, and playgroups. Playgroups and daycare centers are similar in the sense that both are group-based, the difference being that playgroups often do not allow children below the age of 3 and usually offer care for a couple of hours up to half a day and for a limited number of days per week, for example only two or three. Different from a previous analysis [61], the current report focused on external childcare received before Kindergarten only. We therefore measured the average number of days per week of external care across the full period before Kindergarten. On average, the children in the sample entered Kindergarten at age 4.9 years. We also computed squared variables.

## Behavioral development

Outcome variables were measured at ages 7 through 20. S2–S4 Tables show time-points, internal consistencies, and descriptive statistics for all outcome variables.

Social behavior was measured using the Social Behavior Questionnaire (SBQ). The SBQ has shown satisfactory reliability and validity in both the z-proso [70, 71] and other data [72, 73].

*Aggression* included 11 to 12 items per informant (e.g., "The child physically attacks people"). *Attention deficit hyperactivity disorder symptoms* (ADHD symptoms) included 8 to 9 items (e.g., "The child can't sit still, is restless, or hyperactive"), whereas *non-aggressive externalizing behavior* (NAEX; e.g., "The child tells lies and cheats") was measured with 6 to 9 items, *anxiety and depression* (e.g., "The child seems to be unhappy, sad, or depressed") with 7 to 9 items, and *prosocial behavior* (e.g., "The child shares things with peers") with 7 to 10 items per informant.

Responses from the parents and teachers were recorded on a 5-point Likert scale. An easily understandable picture-based yes/no format similar to the Dominique Interactif [74] was used in the first child interviews. Starting age 11, written items were answered on 5-point Likert scales. Scores were z-standardized and averaged. Internal consistencies were acceptable. As usual for cross-behavioral assessments [75], cross-informant reliabilities were low (S5 Table). It is believed that each informant provides incrementally valuable, nonoverlapping information [76]. Because different time-points were available for different informants, we could not combine the scores across informants and instead estimated growth curves for each informant separately.

## Delinquency and substance use

We measured delinquency and substance use in five ways. First, from age 13, the youths reported on their past-year prevalence of 14 types of delinquency (e.g., theft, drug dealing, vandalism, extortion, robbery, assault). We computed a variety score [77].

Second, four self-reported items measured the past-year consumption of tobacco, beer or wine, liquors, and cannabis. Answers were given on a 5-point scale from 1 ("never") to 5 ("daily"). We again computed a variety score. This variable was available from age 13.

Third, we included a broader measure to analyze past-year self-reported deviance at an earlier age (from age 11). Besides delinquency (9 types) and substance use, this variety score included peer aggression (teasing, stealing/destroying possessions, physical violence, and rejection; [78]).

Fourth, the teachers answered seven items on the youths' deviant behavior, including truancy, assault with injury, carrying a knife or weapon, using threats to get something, smoking cigarettes, drinking alcohol, and taking illegal drugs in the past 6 months. We constructed a variety score.

Finally, official delinquency records between ages 10 and 17 were obtained from the Youth Justice Authorities. Over 97% of youths who participated in the study at age 17 provided active consent. We computed a binary variable that indicated whether or not participants had a record and a count measure that assessed the number of trials for which youths had been registered (capped at 2).

## Control variables

We included three types of control variables, all identified a priori, based on prior research, and measured at age 7. The first set concerned characteristics that were known or present at birth, namely the child's sex ('1' signifying males; $M = 0.52$, $SD = 0.50$), single parent household ($M = 0.04$, $SD = 0.21$), maternal age ($M = 30.27$, $SD = 5.38$), prenatal alcohol consumption ($M = 0.46$, $SD = 0.92$), and migrant background ('1' signifying two non-Swiss parents; $M = 0.46$, $SD = 0.50$).

The second set was derived from the EHC and included parental separation (after the child's birth; $M = 0.09$, $SD = 0.28$), parental criminality (i.e., a parent having been a crime suspect; $M = 0.04$, $SD = 0.18$), maternal depression (extended periods of feeling depressed, unhappy, or overburdened; $M = 0.17$, $SD = 0.37$), substantial financial difficulties ($M = 0.13$,

*SD* = 0.33), and parental conflict (periods of serious conflict among caregivers; *M* = 0.16, *SD* = 0.36) before Kindergarten. EHC variables were dichotomous, with '1' indicating that the condition was present.

The third set included variables pertaining to the child's family situation at wave 1 (age 7). Though including these could result in conservative estimates of the associations between childcare and behavior, it is similar to practice in prior studies (e.g., [26]). We included the presence of siblings (*M* = 0.78, *SD* = 0.41), income (*M* = 5.97, *SD* = 1.96), parental education (*M* = 6.40, *SD* = 2.96), and negative parenting (*M* = 1.63, *SD* = 0.29), which included poor monitoring, erratic parenting, and corporal punishment. Correlations between external care and the control variables are shown in S6 Table.

## Analytic strategy

We used growth curve analyses in MPlus [79], using the R packages MplusAutomation [80] and texreg [81]. First, we fitted unconditional models with intercepts and linear slopes. Where enough time-points were available, quadratic and cubic slopes were included and, when significant, retained. Second, we estimated conditional models with childcare and the control variables. For official delinquency, no repeated measures were available, and we used categorical and censored regression. Maximum likelihood with Huber-White covariance adjustment provided robustness to multivariate non-normality and non-independence of observations. We corrected for clustering within schools. To examine the sensitivity of our results to the ratio of the number of clusters to the number of parameters, we also ran models with clustering within classes and without clustering. Results were similar.

In most cases, we used linear models. For binary variables, we estimated logistic models. For self- and teacher-reported delinquency and substance use, we used censored models. MPlus does not provide absolute fit statistics for the latter models and therefore, only the Bayesian Information Criterion (BIC) and the Akaike Information Criterion (AIC) are displayed. Where necessary, non-significant variances and corresponding covariances of the slopes were restricted to zero.

The growth curve models resulted in two types of estimates: relations between external care and the initial level of development at the first available time-point (i.e., the intercept), which was often at age 7, and relations between external care and the rate of change at which children developed after this initial level (i.e., the slope). Time-points for the slope factors were fixed between 0 and 1 to reflect the distance between the measurements. Thus, the slope coefficients represent the overall rate of change from the first to the last time-point included. To test the associations between external childcare and child development at different time-points, additional analyses were performed in which we centered on each time-point available. This is recommended in quadratic and cubic models to provide a more informative analysis [79]. Specifically, the intercept provides the initial status of the growth curve. By setting any time-point to zero (also known as the centering point), the intercept can be interpreted as the level at that particular time-point. Thus, by varying the centering point, we examined the association between external childcare and development at each separate age.

Due to the nature of our sample and data, we could not perform robustness checks to check the sensitivity of our findings against selection bias. For example, change-on-change fixed effects models were not applicable in our case because we analyzed the relation between one prior predictor (external childcare) and a range of later outcomes. Neither could we use sibling fixed effects because with the exception of very rare cases, our data did not include siblings. In addition, propensity score matching requires the matching variables to be measured prior to the main predictor, which was not possible in our data.

We decided against multiple testing and instead guarded against over-interpretation of isolated findings by interpreting our results cautiously, examining overall and consistent patterns in the results rather than isolated findings. Due to space concerns, we refer to S1 Appendix for more information on this, missing data, and other analytic considerations.

The final sample for all analyses reported in this paper consisted of all children whose parents participated in the EHC ($N$ = 1,225; 52% of children was male). The only exception to this were our analyses on the official delinquency records which only included youths who lived in the canton of study at age 17 ($N$ = 939). We used robust full information maximum likelihood to handle missing data on other variables (see S1 Appendix).

Because our data were observational, we could not address causality. For example, issues of reverse causality, with parents enrolling their children into external care due to their behavior, may play a role. We performed exploratory analyses to address this issue. Results, suggesting but not proving that parents did not enroll their children into external care due to their problem behavior, are reported in S2 Appendix.

## Results

### Descriptive analyses

In total, 22% of children received external childcare from family members before Kindergarten, 3% from acquaintances or neighbors, 12% from a daycare mother, 32% in a daycare center, and 22% in a playgroup. Overall, 67% attended some type of external childcare before Kindergarten.

Children generally spent more time in external childcare as they got older (Fig 1). For example, whereas the amount of time children spent in a daycare center in the first year of life was only small, this increased to more than 2.5 days per week on average by age 3 (calculated among children who attended external childcare only). The amount of time children spent in family-based childcare, on the other hand, changed only little as children got older, reaching a peak at about two days a week at age 1 and decreasing slightly thereafter. A bivariate analysis of the relation between (family) characteristics and external childcare is included in S3 Appendix.

### Associations between external childcare and social behavior

The main issue we examined was the association between external childcare and social behavior. Results for the growth curve models are shown by informant in Table 1. All models

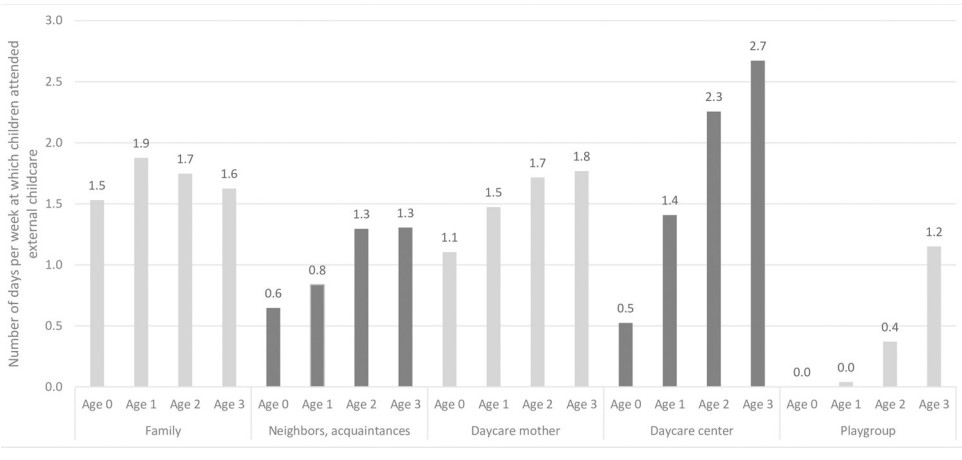

**Fig 1. Number of days per week at which children attended external childcare by age.**

**Table 1. Associations between external childcare and social behavior.** Unstandardized coefficients from growth curve models.

| | Aggression | | | | Non-Aggressive Externalizing | | | ADHD Symptoms | | | Anxiety and Depression | | | Prosocial Behavior | | | |
|---|---|---|---|---|---|---|---|---|---|---|---|---|---|---|---|---|---|
| Informant | Self | Self | Parent | Tea-cher | Self | Parent | Tea-cher | Self | Parent | Tea-cher | Self | Parent | Tea-cher | Self | Self | Parent | Tea-cher |
| Ages | 7–9 | 11–20 | 7–11 | 7–15 | 7–9 | 7–11 | 7–15 | 13–20 | 7–11 | 7–15 | 11–20 | 7–11 | 7–15 | 7–9 | 11–20 | 7–11 | 7–15 |
| **Intercept** | | | | | | | | | | | | | | | | | |
| Family | 0.00 | -0.02 | 0.00 | -0.00 | 0.00 | 0.00 | -0.05 | -0.01 | 0.03 | 0.01 | 0.00 | 0.02 | -0.01 | 0.01* | 0.04* | -0.01 | 0.04 |
| Family sq. | – | – | – | – | – | – | 0.02* | – | – | – | – | – | – | – | – | – | – |
| Acquaintances | 0.01 | 0.02 | -0.02 | 0.04 | 0.00 | -0.02 | 0.06 | 0.26** | 0.20 | 0.10 | -0.00 | 0.03 | -0.03 | 0.01 | 0.11 | 0.02 | 0.07 |
| Acquaintances sq. | – | – | – | – | – | – | – | – | -0.06* | – | – | – | – | – | – | – | – |
| Daycare mother | -0.00 | -0.01 | 0.02 | 0.02 | 0.00 | 0.02 | 0.03 | 0.04 | 0.02 | 0.06 | 0.05 | 0.01 | -0.02 | 0.01 | 0.03 | -0.12** | -0.06 |
| Daycare mother sq. | – | – | – | – | – | – | – | – | – | – | – | – | – | – | – | 0.03* | – |
| Daycare center | 0.02** | -0.02 | 0.02 | 0.05* | 0.00 | 0.03** | 0.03 | 0.02 | 0.04** | 0.06* | 0.02 | 0.02* | -0.01 | 0.00 | 0.00 | 0.01 | -0.01 |
| Playgroup | -0.00 | -0.02 | -0.01 | -0.12 | -0.01 | 0.11* | 0.04 | 0.06 | 0.05 | -0.08 | 0.03 | 0.03 | -0.21* | 0.01 | 0.01 | -0.05 | 0.26* |
| Playgroup sq. | – | – | – | 0.19** | – | -0.07* | – | – | – | – | – | – | 0.20* | – | – | – | -0.23** |
| **Slope** | | | | | | | | | | | | | | | | | |
| Family | -0.00 | 0.12 | -0.02 | -0.16 | -0.00 | -0.02 | 0.13 | -0.12 | -0.00 | 0.01 | 0.03 | -0.02 | 0.02 | -0.01 | 0.12 | 0.00 | 0.05 |
| Family sq. | – | – | – | – | – | – | -0.08 | – | – | – | – | – | – | – | – | – | – |
| Acquaintances | 0.00 | -0.18 | -0.11 | -0.06 | 0.02 | -0.09 | -0.13 | -0.68*** | -0.42** | -0.45 | 0.36* | -0.06 | 0.16* | 0.00 | -0.81 | 0.28 | -0.53 |
| Acquaintances sq. | – | – | – | – | – | – | – | – | 0.10** | – | – | – | – | – | – | – | – |
| Daycare mother | 0.01 | 0.18 | -0.03 | 0.31 | -0.01 | 0.01 | 0.10 | -0.08 | -0.02 | -0.11 | -0.03 | 0.03 | 0.05 | -0.01 | -0.20 | -0.08 | 0.25 |
| Daycare mother sq. | – | – | – | – | – | – | – | – | – | – | – | – | – | – | – | 0.05 | – |
| Daycare center | -0.01** | -0.16 | -0.03 | -0.32* | -0.00 | -0.02 | -0.23** | 0.11 | 0.01 | -0.28 | -0.02 | 0.01 | 0.03 | 0.01 | 0.13 | -0.03 | 0.31 |
| Playgroup | -0.02 | -0.62 | 0.10 | 0.26 | 0.00 | -0.44 | -0.31 | -0.20 | -0.08 | 0.31 | 0.01 | 0.00 | 0.19 | 0.01 | 0.22 | 0.33* | -0.76 |
| Playgroup sq. | – | – | – | -0.80 | – | 0.51 | – | – | – | – | – | – | -0.17 | – | – | – | 0.45 |
| **Quadratic Slope** | | | | | | | | | | | | | | | | | |
| Family | – | 0.18 | 0.01 | 0.35 | – | 0.01 | 0.08 | 0.12 | – | -0.12 | -0.03 | – | – | – | -0.31 | -0.01 | -0.04 |
| Family sq. | – | – | – | – | – | – | -0.02 | – | – | – | – | – | – | – | – | – | – |
| Acquaintances | – | -0.12 | 0.09 | -0.42 | – | 0.08 | -0.05 | 0.36* | – | 0.45 | -0.44** | – | – | – | 1.30 | -0.23 | 1.25 |
| Daycare mother | – | 0.23 | 0.04 | -0.77 | – | -0.03 | 0.28 | 0.06 | – | -0.21 | 0.02 | – | – | – | 0.55 | 0.18 | -0.38 |
| Daycare mother sq. | – | – | – | – | – | – | – | – | – | – | – | – | – | – | – | -0.07 | – |
| Daycare center | – | 0.44 | 0.04 | 0.66* | – | -0.01 | -0.31* | -0.13 | – | 0.39 | 0.02 | – | – | – | -0.41 | 0.03 | -0.74 |
| Playgroup | – | 1.46 | -0.06 | 0.07 | – | 0.26 | -0.60 | 0.08 | – | -1.09 | -0.02 | – | – | – | -0.92 | -0.21 | 2.31 |
| Playgroup sq. | – | – | – | 1.32 | – | -0.39 | – | – | – | – | – | – | – | – | – | – | -1.14 |
| **Cubic Slope** | | | | | | | | | | | | | | | | | |
| Family | – | 0.18 | – | -0.14 | – | – | – | – | – | 0.15 | – | – | – | – | 0.17 | – | -0.05 |
| Family sq. | – | – | – | – | – | – | – | – | – | – | – | – | – | – | – | – | – |
| Acquaintances | – | -0.12 | – | 0.45 | – | – | – | – | – | -0.03 | – | – | – | – | -0.62 | – | -0.73 |
| Daycare mother | – | 0.23 | – | 0.45 | – | – | – | – | – | 0.27 | – | – | – | – | -0.35 | – | 0.14 |
| Daycare mother sq. | – | – | – | – | – | – | – | – | – | – | – | – | – | – | – | – | – |
| Daycare center | – | -0.29 | – | -0.40* | – | – | – | – | – | -0.18 | – | – | – | – | 0.27 | – | 0.43 |
| Playgroup | – | -0.86 | – | -0.29 | – | – | – | – | – | 0.80 | – | – | – | – | 0.73 | – | -1.65 |
| Playgroup sq. | – | – | – | -0.68 | – | – | – | – | – | – | – | – | – | – | – | – | 0.83 |
| $\chi^2$-Value | 35.60 | 59.66 | 52.15 | 164.21 | 37.39 | 41.92 | 150.33 | 52.45 | 38.05 | 123.54 | 37.83 | 55.48 | 185.70 | 25.83 | 47.83 | 57.06 | 174.33 |
| $\chi^2$ df | 20 | 24 | 23 | 100 | 20 | 21 | 100 | 23 | 21 | 96 | 44 | 20 | 145 | 19 | 24 | 24 | 100 |
| CFI | 0.98 | 0.98 | 0.99 | 0.98 | 0.98 | 0.99 | 0.98 | 0.97 | 0.99 | 0.99 | 1.00 | 0.96 | 0.98 | 0.99 | 0.98 | 0.98 | 0.98 |
| TLI | 0.93 | 0.90 | 0.96 | 0.96 | 0.93 | 0.97 | 0.96 | 0.91 | 0.97 | 0.99 | 1.01 | 0.89 | 0.98 | 0.96 | 0.93 | 0.94 | 0.95 |

(*Continued*)

**Table 1.** (Continued)

| Informant | Aggression | | | | Non-Aggressive Externalizing | | | ADHD Symptoms | | | Anxiety and Depression | | | Prosocial Behavior | | | |
|---|---|---|---|---|---|---|---|---|---|---|---|---|---|---|---|---|---|
| | Self | Self | Parent | Tea-cher | Self | Parent | Tea-cher | Self | Parent | Tea-cher | Self | Parent | Tea-cher | Self | Self | Parent | Tea-cher |
| RMSEA Estimate | 0.02 | 0.04 | 0.03 | 0.02 | 0.03 | 0.03 | 0.02 | 0.03 | 0.03 | 0.02 | 0.00 | 0.04 | 0.02 | 0.02 | 0.03 | 0.03 | 0.03 |
| SRMR | 0.01 | 0.01 | 0.01 | 0.02 | 0.01 | 0.01 | 0.02 | 0.01 | 0.01 | 0.01 | 0.01 | 0.01 | 0.02 | 0.01 | 0.01 | 0.01 | 0.01 |
| BIC | 35836.53 | 45128.54 | 41953.37 | 52165.21 | 36172.85 | 41206.06 | 52697.66 | 42264.50 | 45928.93 | 57625.20 | 49107.11 | 43094.43 | 56182.06 | 29778.80 | 42109.18 | 42655.22 | 51818.22 |
| AIC | 34533.30 | 43595.33 | 40542.82 | 50458.23 | 34869.62 | 39657.52 | 50990.69 | 40853.95 | 44508.16 | 56046.00 | 47676.11 | 41791.21 | 54705.07 | 28470.46 | 40575.97 | 41122.01 | 50111.25 |

*** p < 0.001

** p < 0.01

* p < 0.05.

*Notes.* N = 1,225. Sq. = squared. Associations printed in bold are significant at $p < .05$. All covariates included but not shown to avoid clutter. Standard errors not shown to avoid clutter but available from first author.

displayed acceptable fit [82]. An exception was parent-reported anxiety and depression, which had a suboptimal TLI, but other indices were satisfactory. The results under the heading "Intercept" show the association between external childcare and social behavior at the first time-point at which social behavior was measured, which was often age 7. In other words, these models used the age at the first time-point (i.e., often age 7) as the centering point in the growth curve analyses. The results under the heading "Slope" show whether external childcare was associated with changes in social behavior across age. As explained in the Method section, coefficients of quadratic and cubic growth curve models are difficult to interpret. We therefore estimated additional models where we varied the centering point to obtain intercepts for different ages. In other words, we estimated additional models where we examined the relation between external childcare and social behavior at each individual age. These extra analyses are reported in Tables 2–4 as well as in the S1–S3 Appendices. We provide an overview of all results below, by type of childcare.

**Family and acquaintances.** Table 1 shows few significant findings for external childcare by family members and by acquaintances. The models where we varied the centering point (S7 Table) confirmed that there was no consistent evidence that social behavior was related to these two types of external childcare. In order words, we did not find a relation between An exception was that childcare by external family members was related to increased prosocial behavior as reported by the children themselves (ages 7, 11, and 13) and the teachers (ages 8 through 11).

**Daycare mothers.** Table 1 shows no significant findings for external care by daycare mothers. The models where we varied the centering point (Table 2), however, revealed two fairly consistent patterns. First, there were quadratic associations with teacher-reported aggression, NAEX, and ADHD symptoms between ages 9 and 13. The shapes of these associations were similar, and a representative example is found in Fig 2A, indicating that externalizing behavior increased when children spent more than two to three days per week at a daycare mother. Some caution is needed as only 38 children (3%) spent more than two days per week with a daycare mother.

The second pattern of findings occurred only twice and at a fairly late age: a quadratic relation between care by a daycare mother and self-reported aggression at ages 15 and 17. These relations were similar to those for teacher-reported social behavior (see Fig 2B for an example): aggression increased when children spent more than about three days per week at a daycare mother. Though consistent with our previous results, this finding seemed weak and unstable as it did not occur at other ages or for other types of self-reported behavior.

**Daycare centers.** Next, we examined the associations for daycare centers. Table 1 shows that daycare center attendance was related to several types of problem behavior at age 7 (intercept) and also to changes in problem behavior after age 7 (under "Slope", "Quadratic Slope", and "Cubic Slope"). The models where we varied the centering point (Table 3) provide easier interpretation. First, the findings in Table 3 show that daycare center care was associated with increased parent-reported aggression at ages 9 and 11, NAEX at ages 7 and 9, ADHD symptoms at ages 7 to 11, and anxiety and depression at ages 7 to 11. Three associations were quadratic, showing a weak negative relation (see Fig 2C for a representative example). All other associations were linear, suggesting that more time in a daycare center was associated with increased problem behavior.

Other results were less consistent. For example, daycare center care was associated with more self-reported aggression at ages 7 and 8, but the relation flipped at ages 13 and 15, where center care was related to less aggression. Because this latter finding was somewhat unexpected given our other findings, we examined it further and performed robustness checks, which are reported in S4 Table. More time in a daycare center was also associated with more self-

**Table 2. Amount of time in care by <u>daycare mothers</u> and social behavior by age.** Unstandardized coefficients from growth curve models.

| Approx. age | 7 | 8 | 9 | 10 | 11 | 12 | 13 | 15 | 17 | 20 |
|---|---|---|---|---|---|---|---|---|---|---|
| **PARENT REPORTS** | | | | | | | | | | |
| Aggression | n. s. | n. s. | n. s. | | **b = .03** | | | | | |
| Non-aggressive externalizing | n. s. | n. s. | n. s. | | n. s. | | | | | |
| ADHD symptoms | n. s. | | n. s. | | n. s. | | | | | |
| Anxiety and depression | n. s. | | n. s. | | n. s. | | | | | |
| Prosocial behavior | **b = -.12, c = 0.03** | **b = -0.13, c = 0.04** | **b = -0.12, c = 0.04** | | n. s. | | | | | |
| **SELF REPORTS** | | | | | | | | | | |
| Aggression | n. s. | n. s. | n. s. | | n. s. | | n. s. | **b = -.11, c = .04** | **b = -.10, c = .03** | n. s. |
| Non-aggressive externalizing | n. s. | n. s. | n. s. | | | | | | | |
| ADHD symptoms | | | | | | | n. s. | n. s. | n. s. | n. s. |
| Anxiety and depression | | | | | n. s. | | n. s. | n. s. | n. s. | n. s. |
| Prosocial behavior | n. s. | n. s. | n. s. | | n. s. | | n. s. | n. s. | n. s. | n. s. |
| **TEACHER REPORTS** | | | | | | | | | | |
| Aggression | n. s. | n. s. | n. s. | b = -.09, c = .04 | **b = -.11, c = .04** | **b = -.11, c = .04** | **b = -.11, c = .03** | n. s. | | |
| Non-aggressive externalizing | n. s. | n. s. | b = -.04, c = .02 | b = -.06, c = .02 | **b = -.08, c = .03** | **b = -.09, c = .03** | **b = -.08, c = .02** | n. s. | | |
| ADHD symptoms | n. s. | n. s. | n. s. | n. s. | **b = -.18, c = .05** | **b = -.19, c = .05** | **b = -.17, c = .04** | n. s. | | |
| Anxiety and depression | n. s. | n. s. | n. s. | n. s. | n. s. | n. s. | n. s. | n. s. | | |
| Prosocial behavior | n. s. | n. s. | n. s. | n. s. | n. s. | n. s. | n. s. | n. s. | | |

*Notes*. N = 1,225. Associations printed in bold are significant at $p < .05$. n. s. = not significant. b = unstandardized coefficient. c = unstandardized quadratic coefficient. Gray boxes: outcome measures not available. All covariates included but not shown to avoid clutter.

reported ADHD symptoms at ages 15 and 17, though not at other ages. There was some evidence that more time spent in a daycare center was associated with teacher-reported aggression and ADHD symptoms, but this relation disappeared after age 7.

**Playgroups.** Finally, we examined the associations for playgroups. Table 1 shows several relations between playgroup attendance and the initial level of social behavior (under "Intercept") and changes in social behavior thereafter (under "Slope", "Quadratic Slope", and "Cubic Slope"). Results of the models where we varied the centering point are shown in Table 4. There were three overall findings with varying consistency. First, there were inconsistent associations

**Table 3. Amount of time spent in a <u>daycare center</u> and social behavior by age.** Unstandardized coefficients from growth curve models.

| Approx. age | 7 | 8 | 9 | 10 | 11 | 12 | 13 | 15 | 17 | 20 |
|---|---|---|---|---|---|---|---|---|---|---|
| **PARENT REPORTS** | | | | | | | | | | |
| Aggression | n. s. | n. s. | b = .06, c = -.02 | | b = .03 | | | | | |
| Non-aggressive externalizing | b = .03 | b = .07, c = -.02 | b = .07, c = -.02 | | n. s. | | | | | |
| ADHD symptoms | b = .04 | | b = .05 | | b = .06 | | | | | |
| Anxiety and depression | b = .02 | | b = .03 | | b = .03 | | | | | |
| Prosocial behavior | n. s. | n. s. | n. s. | | n. s. | | | | | |
| **SELF REPORTS** | | | | | | | | | | |
| Aggression | b = .02 | b = .01 | n. s. | | n. s. | | b = -.03 | b = -.03 | n. s. | n. s. |
| Non-aggressive externalizing | n. s. | n. s. | n. s. | | | | | | | |
| ADHD symptoms | | | | | | | n. s. | b = .04 | b = .05 | n. s. |
| Anxiety and depression | | | | | n. s. | | n. s. | n. s. | n. s. | n. s. |
| Prosocial behavior | n. s. | n. s. | n. s. | | n. s. | | n. s. | n. s. | b = -0.10, c = 0.04 | n. s. |
| **TEACHER REPORTS** | | | | | | | | | | |
| Aggression | b = .05 | n. s. | n. s. | n. s. | n. s. | n. s. | n. s. | n. s. | | |
| Non-aggressive externalizing | n. s. | n. s. | n. s. | n. s. | n. s. | n. s. | n. s. | n. s. | | |
| ADHD symptoms | b = .06 | n. s. | n. s. | n. s. | n. s. | n. s. | n. s. | n. s. | | |
| Anxiety and depression | n. s. | n. s. | n. s. | n. s. | n. s. | n. s. | n. s. | n. s. | | |
| Prosocial behavior | n. s. | n. s. | n. s. | n. s. | n. s. | n. s. | n. s. | n. s. | | |

*Notes*. N = 1,225. Associations printed in bold are significant at $p < .05$. n. s. = not significant. b = unstandardized coefficient. c = unstandardized quadratic coefficient. Gray boxes: outcome measures not available. All covariates included but not shown to avoid clutter.

for parent-reported child behavior. Though the quadratic association at age 7 suggested that attending a playgroup for more than two days a week was associated with decreasing NAEX (Fig 2D), this was not the case at age 8, where more hours in a playgroup linearly related to more NAEX. Null findings dominated the associations for other ages and other parent-reported behavior.

Second, playgroups were associated with lower self-reported aggression at ages 13 and 15. At age 17, the relation was quadratic and also suggested that time spent in a playgroup was associated with lower aggression (Fig 2E). At earlier ages and for other types of self-reported social behavior, results were dominated by null findings.

Third, the most consistent results were found for teacher-reported aggression, anxiety and depression, and prosocial behavior. All associations were quadratic and similar in shape (see Fig 2F–2H for representative examples), suggesting that attending a playgroup on at least 1.5 days was associated with more aggression at ages 7 and 8, more anxiety and depression at ages 7 through

**Table 4. Relations between amount of time spent in a <u>playgroup</u> and social behavior by age.** Unstandardized coefficients from growth curve models.

| Approx. age | 7 | 8 | 9 | 10 | 11 | 12 | 13 | 15 | 17 | 20 |
|---|---|---|---|---|---|---|---|---|---|---|
| **PARENT REPORTS** | | | | | | | | | | |
| Aggression | n. s. | n. s. | n. s. | | n. s. | | | | | |
| Non-aggressive externalizing | **b = .11, c = -.07** | **b = .06** | n. s. | | n. s. | | | | | |
| ADHD symptoms | n. s. | | n. s. | | n. s. | | | | | |
| Anxiety and depression | n. s. | | n. s. | | n. s. | | | | | |
| Prosocial behavior | n. s. | n. s. | n. s. | | n. s. | | | | | |
| **SELF REPORTS** | | | | | | | | | | |
| Aggression | n. s. | n. s. | n. s. | | n. s. | | **b = -.10** | **b = -.08** | **b = .04, c = -.06** | n. s. |
| Non-aggressive externalizing | n. s. | n. s. | n. s. | | | | | | | |
| ADHD symptoms | | | | | | | n. s. | n. s. | n. s. | n. s. |
| Anxiety and depression | | | | | n. s. | | n. s. | n. s. | n. s. | n. s. |
| Prosocial behavior | n. s. | n. s. | n. s. | | n. s. | | n. s. | n. s. | n. s. | n. s. |
| **TEACHER REPORTS** | | | | | | | | | | |
| Aggression | b = -.12, **c = .19** | b = -.09, **c = .12** | n. s. | n. s. | n. s. | n. s. | n. s. | n. s. | | |
| Non-aggressive externalizing | n. s. | n. s. | n. s. | n. s. | n. s. | n. s. | n. s. | n. s. | | |
| ADHD symptoms | n. s. | n. s. | n. s. | n. s. | n. s. | n. s. | n. s. | n. s. | | |
| Anxiety and depression | **b = -.21, c = .20** | **b = -.20, c = .18** | **b = -.17, c = .16** | **b = -.14, c = .13** | **b = -.12, c = .12** | n. s. | n. s. | n. s. | | |
| Prosocial behavior | **b = 0.26, c = -0.23** | **b = 0.21, c = -0.20** | **b = 0.19, c = -0.18** | **b = 0.22, c = -0.18** | **b = 0.25, c = -0.19** | **b = 0.29, c = -0.19** | **b = 0.29, c = -0.18** | n. s. | | |

*Notes*. N = 1,225. Associations printed in bold are significant at p < .05. n. s. = not significant. b = unstandardized coefficient. c = unstandardized quadratic coefficient. Gray boxes: outcome measures not available. All covariates included but not shown to avoid clutter.

11, and less prosocial behavior at ages 7 through 13. Since children generally do not attend playgroups before age 3, this means in practice that the cut-off was around 3 days per week.

Summing up, evidence suggested that daycare center attendance was related to more problem behavior according to the parents and, to some extent, the children themselves. Spending more than two to three days per week at a daycare mother and visiting a playgroup on at least three days per week was related to more externalizing and internalizing behavior and less prosocial behavior according to the teachers. Relations decreased as the children got older.

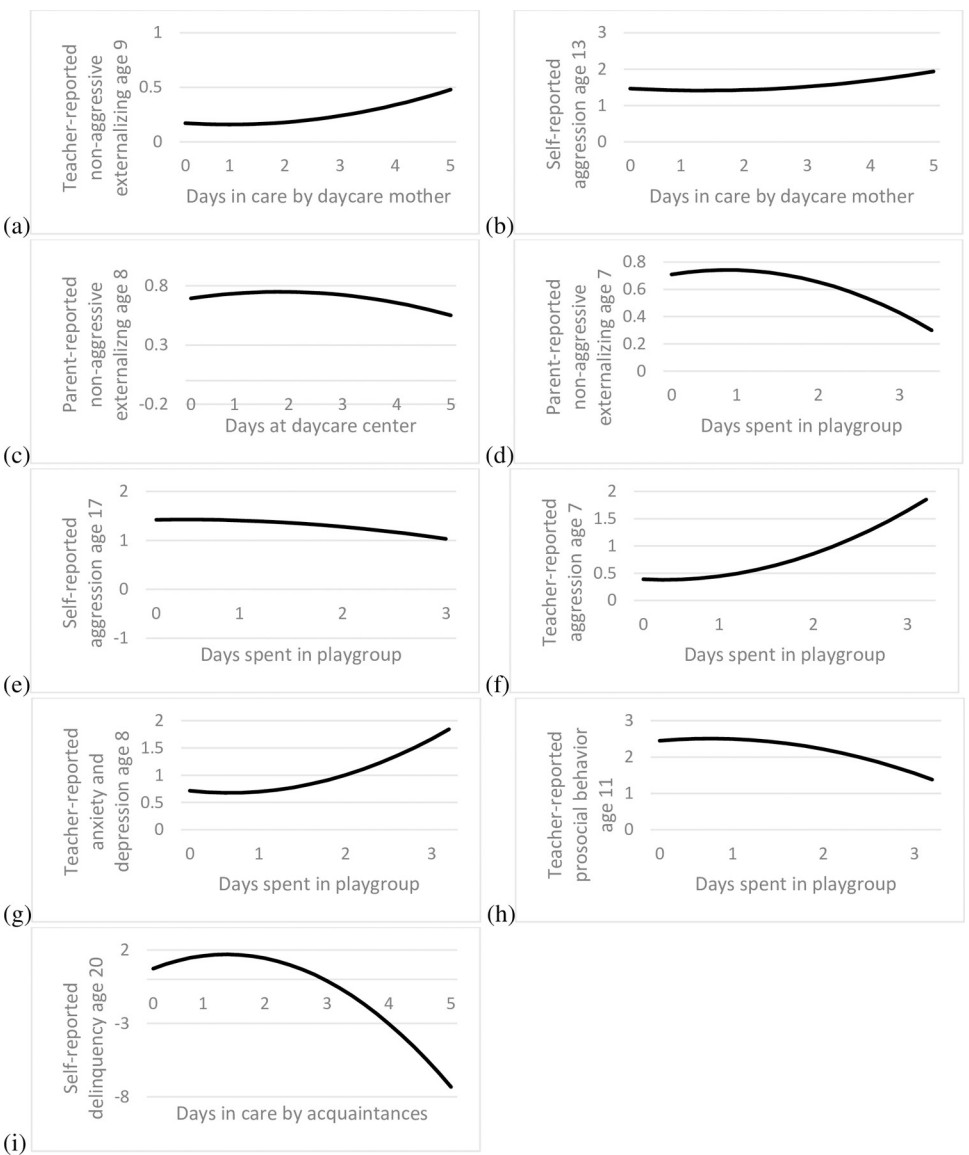

**Fig 2. Examples of curvilinear relations between time spent in external childcare and social behavior.**

## Associations between external childcare, delinquency, and substance use

We proceeded with growth curve analyses for the relation of external care with delinquency and substance use. Results are shown in Table 5. These models used early adolescence (age 10, 11, or 13, depending on the earliest availability; see Table 5) as the centering point, so the intercept is defined at this age. We again estimated additional models where we varied the centering point to obtain intercepts for different ages and interpret the coefficients. All results are discussed below.

**Family and daycare mothers.** Table 5 shows no significant findings for external care by family members and daycare mothers. This was confirmed in the models where we varied the centering point (S10 and S12 Tables), where few if any consistent findings emerged.

**Acquaintances.** Table 5 shows that care by acquaintances was related to more self-reported delinquency and substance use at age 13 and deviance at age 11 (under "Intercept").

**Table 5. Associations between external childcare, delinquency, and substance use.** Unstandardized coefficients from growth curve models.

| | Delinquency | Deviance | Substance Use | Delinquency and substance use |
|---|---|---|---|---|
| **Informant** | **Self** | **Self** | **Self** | **Teacher** |
| **Ages** | **Ages 13–20** | **Ages 11–20** | **Ages 13–20** | **Ages 10–15** |
| **Intercept** | | | | |
| Family | 0.12 | -0.01 | 0.06 | -0.01 |
| Acquaintances | 0.57 | **0.53**\* | **0.28**\* | 0.08 |
| Daycare mother | -0.02 | -0.03 | -0.07 | -0.01 |
| Daycare center | 0.16 | 0.03 | -0.08 | -0.05 |
| Daycare center sq. | – | – | **0.07**\* | – |
| Playgroup | -0.57 | -0.78 | **-0.43**\* | 0.09 |
| Playgroup sq. | – | **0.44**\* | **0.33**\* | – |
| **Slope** | | | | |
| Family | 0.01 | -0.09 | -0.30 | -0.30 |
| Acquaintances | -3.90 | -0.46 | -0.25 | -0.48 |
| Daycare mother | -0.48 | -0.01 | 0.35 | 0.28 |
| Daycare center | -0.81 | 0.02 | **0.78**\*\* | 0.04 |
| Daycare center sq. | – | – | **-0.39**\*\*\* | – |
| Playgroup | 0.80 | **0.97**\* | **1.60**\* | -0.01 |
| Playgroup sq. | – | **-0.87**\*\*\* | **-1.09**\* | – |
| **Quadratic Slope** | | | | |
| Family | -0.30 | – | 0.23 | 0.33 |
| Acquaintances | 3.09 | – | 0.12 | 0.90 |
| Daycare mother | 0.42 | – | -0.28 | -0.42 |
| Daycare center | 0.80 | – | **-0.62**\*\* | -0.22 |
| Daycare center sq. | – | – | **0.31**\*\*\* | – |
| Playgroup | -0.04 | – | -1.11 | -0.11 |
| Playgroup sq. | – | – | 0.68 | – |
| $\chi^2$-Value | – | – | 24.42 | – |
| $\chi^2$ df | – | – | 22 | – |
| CFI | – | – | 1.00 | – |
| TLI | – | – | 0.99 | – |
| RMSEA Estimate | – | – | 0.01 | – |
| SRMR | – | – | 0.01 | – |
| BIC | 44845.04 | 57308.44 | 50772.95 | 39039.89 |
| AIC | 43408.94 | 55877.45 | 49096.64 | 37608.90 |

\*\*\*$p < 0.001$

\*\*$p < 0.01$

\*$p < 0.05$.

*Notes*. N = 1,225. Sq. = squared. Associations printed in bold are significant at $p < .05$. All covariates included but not shown to avoid clutter.

The additional models where we varied the centering point help interpret the significant findings under "Slope" and "Quadratic Slope". The results of these models (S11 Table) revealed five significant relations. Four of these showed that external care by acquaintances was related to more self- and teacher-reported delinquency and substance use at ages 11, 13, and 15, though not at ages 10, 12, and 17. The fifth significant effect was on self-reported delinquency at age 20, which was a quadratic relation. Plotting this relation (see Fig 2I) revealed that

spending more than, roughly, 2.5 days in this type of external childcare was related to lower delinquency, which was not consistent with the other four findings and seemed an isolated result.

**Daycare centers and playgroups.**  Table 5 shows several quadratic relations for daycare centers and playgroups. We examined these relations by varying the centering point (S13 and S14 Tables) but found that most results showed null effects. Thus, there was little consistent evidence that daycare centers and playgroups were related to delinquency and substance use.

As an alternative measure of delinquency, we examined official delinquency records in relation to external childcare. There were no significant relations (S15 Table).

Summing up, we found little consistent evidence for a relation between external childcare, delinquency, and substance use in adolescence.

## Are the relations different for children from vulnerable backgrounds?

Finally, we examined relations between vulnerability, external childcare, and behavior. Similar to prior studies (e.g., [43]), we computed a cumulative risk score to measure vulnerability. From the fourteen control variables that were included in the other analyses reported in this paper, twelve (excluding sex and having siblings) were selected and summed. Non-dichotomous variables were cut off at 1 standard deviation worse than the mean. Thus, the final risk score reflected the number of risk factors that were present in a child's background. We capped the risk score at six to avoid extreme outliers and multiplied childcare by the risk score (after centering both). We then entered the interactions and main effects into the analyses.

Besides the overall growth curve analyses with the earliest available age as the centering points (S16 Table), we again estimated models where we varied the centering point. Due to space limitations, all tables with results are shown in the Online Appendix. We discuss the results below by type of external childcare and provide plots to interpret the significant findings.

**Family members and playgroups.**  We found no evidence for a relation between external care, risk, and behavior for childcare by family members and in playgroups (S17 and S21 Tables). In other words, childcare by family members and in playgroups was mostly unrelated to children's later social behavior for both children from more and less vulnerable backgrounds.

**Acquaintances and daycare mothers.**  Some evidence suggested relations between external childcare by acquaintances, risk, and (mostly teacher-reported) social behavior (S18 Table). Also, some evidence suggested relations for external childcare by daycare mothers (S19 Table). Although these were relatively isolated findings, plotting these interaction terms suggested some consistency (see Fig 3A and 3B for representative examples), with more care by acquaintances and by daycare mothers being associated with more externalizing and less prosocial behavior for children from high-risk backgrounds, and with less externalizing and more prosocial behavior for children from low-risk backgrounds. The relation flipped in adolescence (Fig 3C); however, this finding was even more isolated.

**Daycare centers.**  Regarding external childcare in daycare centers, there was some evidence for a relation between daycare attendance, risk, and (primarily teacher-reported) social behavior (S20 Table). Although most of the relations were not statistically significant, plots suggested some consistency in the significant results (see Fig 3D for an example): For children from vulnerable backgrounds, more time in a daycare center was related to less externalizing behavior. The same was not the case for children from less vulnerable backgrounds, for whom results showed no relations or for whom more time in a daycare center was related to more externalizing behavior. In addition, there were some relations between care in daycare centers,

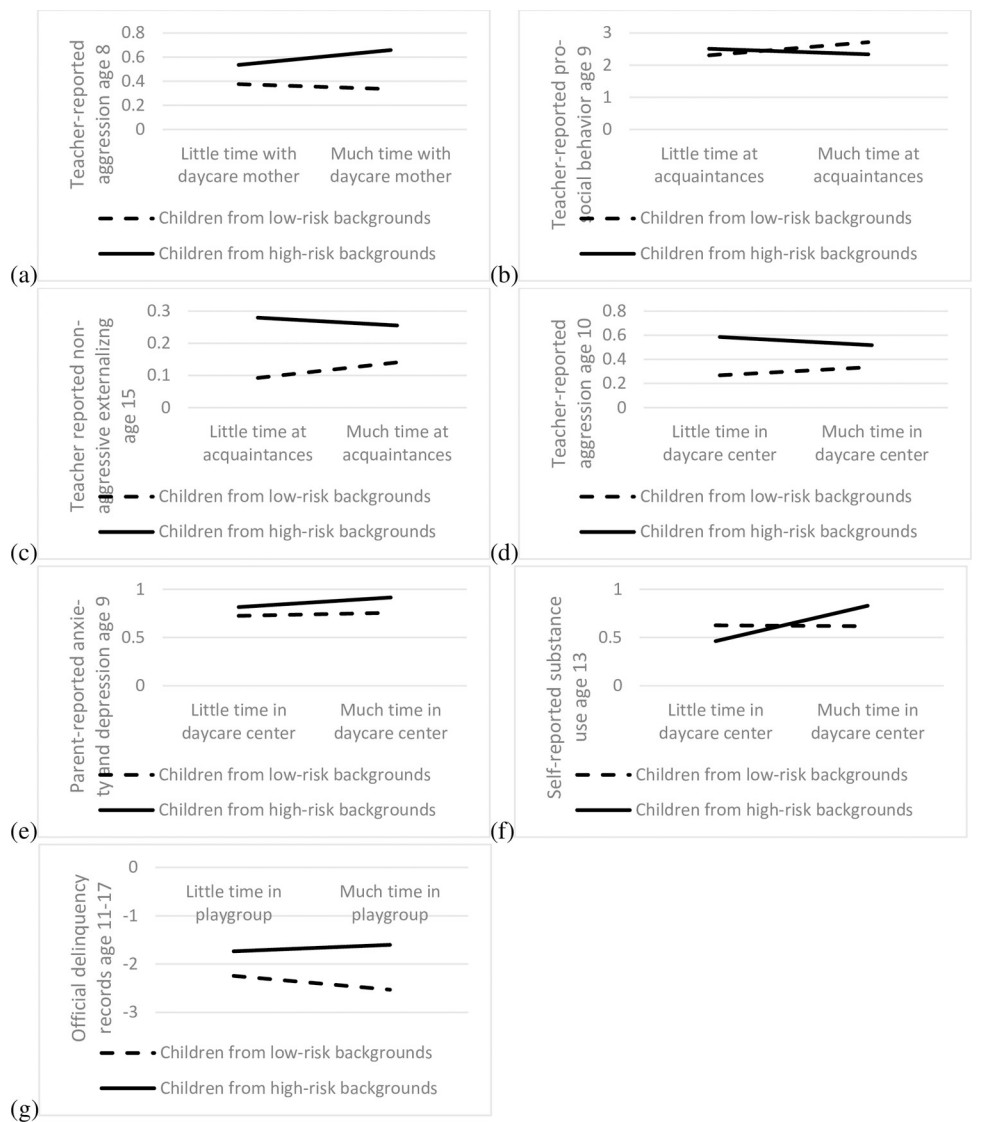

**Fig 3. Examples of interactions between external childcare, risk, and social behavior.**

risk, and internalizing behavior. Though these were more isolated, plotting them revealed consistency in their direction (see Fig 3E for an example): For children from vulnerable backgrounds, spending more time in a daycare center was related to more internalizing problems.

## The relation between external childcare, risk, delinquency, and substance use

We repeated our analyses on the relation between external childcare and risk for delinquency and substance use. In other words, we examined whether our earlier findings that external childcare was mostly unrelated to delinquency and substance use applied to children from both more and less vulnerable backgrounds (S22 Table). In line with our earlier procedure, we estimated growth curve models and additional models where we varied the centering points. The results were dominated by null findings with one consistent exception: The interaction between daycare center attendance and risk was significantly related to substance use at ages

**Table 6. Overview of main findings on the relation between external childcare and socio-behavioral development by type of care.**

| Type of external childcare | Results by child development domain |
| --- | --- |
| External family members | **Social behavior**: Childcare by external family members was related to increased prosocial behavior as reported by the children (ages 7, 11, and 13) and the teachers (ages 8 through 11). |
| | **Delinquency and substance use**: No consistent evidence for a relation. |
| Acquaintances | **Social behavior**: Inconsistent findings. |
| | **Delinquency and substance use**: Some evidence suggested that external care by acquaintances was related to more delinquency and substance use in adolescence. |
| Daycare mother | **Social behavior**: Spending more than, roughly, 3 days per week at a daycare mother before Kindergarten was related to more externalizing behavior (aggression, non-aggressive externalizing, and ADHD symptoms) at ages 9 to 13 according to the teachers. |
| | **Delinquency and substance use**: No consistent evidence for a relation. |
| Daycare center | **Social behavior**: Spending more time in a daycare center was related to more externalizing behavior and internalizing problems according to the parents. This was the case until at least age 11, when the parent interviews were ceased. Some evidence suggested that more time spent in a daycare center was related to more ADHD symptoms into adolescence (ages 15 to 17) according to the youths themselves. |
| | There was some evidence that children's background mattered. Specifically, there was some evidence that for children from more vulnerable backgrounds, attending a daycare center was associated with less externalizing behavior and more internalizing problems. For children from less vulnerable backgrounds, however, attending a daycare center was associated with more externalizing behavior. |
| | **Delinquency and substance use**: No consistent evidence for a relation with delinquency. For children from vulnerable backgrounds, spending more time in a daycare center was related to more substance use in adolescence. |
| Playgroup | **Social behavior**: Visiting a playgroup on more than, roughly, three days per week was related to more aggression at ages 7 and 8 and higher anxiety and depression at ages 7 through 11 according to the teachers. |
| | **Delinquency and substance use**: No consistent evidence for a relation. |

13, 15, 17, and 20. All interaction terms (see Fig 3F for a representative example) showed the same pattern: more time in a daycare center was related to more substance use for youths from vulnerable backgrounds but not for youths from less vulnerable backgrounds.

In a final step, we examined the relation between external care, risk, and the official criminal records (S23 Table). There was one significant interaction, namely between playgroup attendance and risk on the incidence of delinquency. It is displayed in Fig 3G and similar to the interaction between substance use and risk: more time in a playgroup was related to a higher risk of having official delinquency records for children from high-risk backgrounds, but to a lower risk of having official delinquency records for children from low-risk backgrounds. This finding seems isolated and was not replicated for the binary delinquency measure, however.

In order to provide a concise summary of the study's main findings, an overview is included in Table 6.

## Discussion

We examined the relation of external childcare before Kindergarten with child and youth development from ages 7 to 20. Our study was novel because to our knowledge, no prior study has examined a similarly wide range of social behavior, external care arrangements, and particularly length of follow-up. Our findings showed that external childcare was related to social behavior in a number of ways, but that the type of childcare, the type of social behavior, age, and informant mattered.

For externalizing behavior, we found that spending more time in a daycare center was related to more problem behavior according to the parents. The relations persisted until at least age 11 (when the parent interviews were ceased). Some evidence suggested that it was also related to more ADHD symptoms in adolescence (age 15 and 17) as reported by the youths themselves. More than, roughly, three days per week at a daycare mother or in a playgroup across the period before Kindergarten was related to increased teacher-reported aggression, non-aggressive externalizing behavior, and ADHD symptoms until age 11 to 13. We found some relations between external childcare by acquaintances and behavior, but these were inconsistent, which may be due to the small number of children who received this type of external childcare (only 3%).

Our findings on externalizing behavior are in line with some, though not all, of the very few studies that examined long-term relations, which have found that the relation between time spent in a daycare center and externalizing problems may persist until the end of primary school or into adolescence ([20, 41], but see [44, 51]). On the other hand, our findings on day-care mothers and playgroups are more difficult to situate in the context of prior literature, because the vast majority of prior studies has not specifically focused on these types of external childcare.

Regarding delinquency, our findings suggested little consistent evidence for a relation with external childcare. Few studies have been done on the relation between external childcare and delinquency. One exception was the study by Baker et al. [50] in Quebec, Canada, which found that external childcare was associated with a 20% higher crime rate at ages 12 to 20. Our findings did not replicate this but instead were in line with our other results, which indicated that the relations between external childcare and behavior weakened as children moved into adolescence.

In addition to our findings on externalizing behavior, we also found that spending more time in a daycare center was related to more anxiety and depression at ages 7 to 11. This find-ing is in line with those from some prior studies [28] but not others (e.g., [22, 34]). The inter-national literature has been mixed regarding the relation between external childcare and internalizing problems. One explanation for this could be that the findings depend on chil-dren's background, with some children profiting less or more from external childcare than others. We elaborate on this further below.

On the positive side, we found evidence that care by family members was associated with increased prosocial behavior, which to our knowledge has not been examined or found before. Possibly, being embedded in a larger family context can provide children with the love, belong-ing, and trust [83] needed to promote positive behavior. The other types of external childcare might not provide these baseline conditions as they were not related to prosocial behavior.

Finally, we found no strong evidence that children from more vulnerable backgrounds profited more from external childcare compared to other children. There were some notewor-thy findings, however. First, care by acquaintances and by daycare mothers was associated with more externalizing behavior for children from vulnerable backgrounds and less external-izing behavior for children from less vulnerable backgrounds. This was somewhat surprising as we are not aware of prior research or theory that might explain this. Possibly, parents from vulnerable backgrounds are more likely to choose acquaintances and daycare mothers who are also in vulnerable circumstances, potentially adding to the child's stress and promoting prob-lem behavior.

Second, some of our findings showed that spending more time in a daycare center was related to less externalizing behavior for children from vulnerable backgrounds, but to more problem behavior for children from less vulnerable backgrounds. Although the international literature has been mixed on this topic, our findings are in line with some prior studies that,

like ours, used a cumulative risk index [43]. A structured center-based environment may help children from high-risk backgrounds to control their externalizing behavior. Children from low-risk backgrounds, on the other hand, might experience more aggression in center-based childcare than at home and imitate their peers' problem behavior [32].

On the other hand, our data uniquely suggest that daycare center care was related to both more internalizing problems and substance use for children from vulnerable backgrounds. For substance use, the relation persisted through adolescence and even into early adulthood. Thus, the structured context and rules in a daycare center may help these young people to control their externalizing behavior but not their internalizing problems and substance use. It is possible that the experienced control and suppression of externalizing behavior leads to internalizing symptoms. Alternatively, children from precarious conditions may experience daycare centers as more unpleasant and hostile than other children, because daycare centers (like non-precarious households) have certain "middle class" behavioral norms, which could lead to a permanent feeling of alienation. Furthermore, in line with the bioecological model [2], these children might experience inner conflict because the behavioral norms in their home environment differ from those at the daycare center.

One interesting finding is that our results differed by informant. Whereas external childcare in a daycare center was related to externalizing behavior according to the parents (and, to some extent, the children themselves), it was external childcare by a daycare mother or in a playgroup that was related to problem behavior according to the teachers. One potential explanation may be that informants observe different contexts. Daycare centers are probably most similar to the school environment, which may be why teachers do not observe effects of daycare centers. Prior research to some extent supports this argument, finding that both daycare centers and schools increased children's level of stress (i.e., cortisol) and that stress levels at school were just as high for children who had attended a daycare center and those who had not [84]. External care by daycare mothers, on the other hand, takes place in a home environment, which may be why parents notice fewer differences between children who did and did not visit a daycare mother than teachers. Playgroups, finally, are more integrated into home-based care than daycare centers, which may be why parents notice differences in behavior less often than teachers. Our findings highlight the importance of multi-informant data when examining external childcare.

Due to the wide range of outcomes, types of childcare, and time-span included, as well as the multi-informant nature of our data, we were able to paint a nuanced picture. Though there were some results suggesting relations between childcare and productive youth development (e.g., the association between external family care and prosocial behavior, and decreased externalizing behavior among vulnerable children in daycare), other results suggested relations between childcare and increased problem behavior (e.g., the association between daycare and externalizing, and increased internalizing problems and substance use among vulnerable children in daycare), highlighting the necessity of examining various outcome domains. All in all, our findings suggest that early life experiences in external care shape children's pathways, suggesting that a life course view is important in the relation between external childcare and development. Ultimately, other contexts and experiences gain precedence, and most of the relations diminished as children got older, although some remained until late adolescence and early adulthood.

We cannot claim that our results are causal because of limitations of our data. First, as described in the introduction, studies that examine the relation between external childcare and child development cannot rule out selection bias, and our study is no exception. That is, whether or not children attend external childcare is related to characteristics of the children and their families. In turn, child development is related to these characteristics, too, making it

unclear whether differences in child development between children can be attributed to external childcare or to other characteristics. Like many previous studies, we have tried our best to reduce this problem by including a range of control variables that may explain why children entered external childcare. However, our data and statistical design do not allow us to rule out other potential explanations. For example, we did not know whether or not parents enrolled their children in external childcare because they displayed problem behavior. As mentioned, we performed robustness analyses to examine how likely it is that our results are influenced by this (see S3 Appendix). Even though prior research has shown that the most often cited reasons for parents to enroll their children in external childcare is that the parents were working [20], we cannot rule out the possibility that their children's problem played a role. In addition, we did not have information on negative parenting, maternal work, parents' personality characteristics, or the parents' socio-economic status *before* their children attended external childcare. Instead, we only had this information at the time of the first interviews, which took place when the children were seven years old. Thus, we could not examine whether external childcare affected these family characteristics. Summing up, even though we included a range of control variables in our analyses, we cannot rule out the possibility that our results are due to other, unmeasured, characteristics of children and families that may explain the relation between external childcare and child development.

As another limitation, we did not have data on the quality of external childcare, which varies widely in Switzerland [85]. International studies have found some support for the notion that children visiting very high quality centers showed less externalizing behavior than those visiting low or regular quality centers (e.g., [5]). However, the previously mentioned NICHD study, which used a multisite longitudinal design in the US and regression analysis with covariate adjustment, found that even though low-quality external care is more often associated with problem behavior than high-quality care, external care was related to more externalizing behavior at all levels of quality [6].

In addition, information on external childcare was collected retrospectively, at age 7. Although the reliability of retrospective childcare reports is supported [1990], the EHC used in our study has shown validity [68, 69], and all data were collected with care, prior research has shown that people generally underestimate the frequency of life events [86]. This may be due to memory effects, but motivation issues may also be a source of bias [87]. Though we are not aware of studies on retrospective bias in reports of childcare, we cannot rule out the possibility of bias in these retrospective self-reported data.

Also, because of the nature of our longitudinal data, our measures on external care were from the years 1997 to 2002. Since then, the provision of external childcare has seen a rapid expansion. Although the city promotes the quality of care, to our knowledge no data exist on how daycare centers in Switzerland have changed. In the Netherlands, which has seen a similar expansion of external childcare, the increase in the quantity of daycare centers was accompanied by a decline in quality [88]. It is unclear whether the same applies to Switzerland.

Finally, due to space constraints, we could not examine differences by gender [51]. The question whether gender moderated the relations remains a question for future research.

Although there were limitations to our study, to our knowledge there are no other Swiss studies with comparably extensive data that can be linked to external childcare. Even internationally, few if any similar studies exist. Summing up, our study found that external care tends to be related to increased externalizing behavior (i.e., aggression, non-aggressive externalizing, and ADHD symptoms), internalizing problems (i.e., anxiety and depression), and substance use, but that most though not all relations diminished and disappeared as children grew into adolescents and young adults. An exception was the relation between daycare attendance and substance use, which persisted into young adulthood for those from vulnerable backgrounds.

## Supporting information

**S1 Table. Participation and retention rates.**
(DOCX)

**S2 Table. Time-points at which included outcome variables were measured.**
(DOCX)

**S3 Table. Internal consistency (and number of items) for the Social Behavior Questionnaire by time-point and informant (Cronbach's alpha).**
(DOCX)

**S4 Table. Descriptive statistics for the Social Behavior Questionnaire by time-point and informant (mean (standard deviation)).**
(DOCX)

**S5 Table. Cross-informant reliabilities for the Social Behavior Questionnaire (Cronbach's alpha).**
(DOCX)

**S6 Table. Bivariate correlations of types of external childcare with control variables.**
(DOCX)

**S7 Table. Relations between the amount of time spent in external care by family members and social behavior by age.** Unstandardized coefficients from growth curve models.
(DOCX)

**S8 Table. Relations between amount of time spent in external care by acquaintances and social behavior by age.** Unstandardized coefficients from growth curve models.
(DOCX)

**S9 Table.** a. Correlations between externalizing behavior and internalizing problems at ages 0 to 1 and enrollment in external childcare at age 2. b. Correlations between externalizing behavior and internalizing problems at ages 0 to 2 and enrollment in external childcare at age 3.
(DOCX)

**S10 Table. Relations between amount of time spent in external care by family members and deviant behavior by age.** Unstandardized coefficients from growth curve models.
(DOCX)

**S11 Table. Relations between amount of time spent in external care by acquaintances and deviant behavior by age.** Unstandardized coefficients from growth curve models.
(DOCX)

**S12 Table. Relations between amount of time spent in external care by daycare mothers and deviant behavior by age.** Unstandardized coefficients from growth curve models
(DOCX)

**S13 Table. Relations between amount of time spent in a daycare center and deviant behavior by age.** Unstandardized coefficients from growth curve models.
(DOCX)

**S14 Table. Relations between amount of time spent in a playgroup and deviant behavior by age.** Unstandardized coefficients from growth curve models.
(DOCX)

**S15 Table. Relations between external delinquency and official measures of delinquency.** Unstandardized coefficients from regression models. (DOCX)

**S16 Table. Interaction effects between external childcare and risk on social behavior.** Unstandardized coefficients from growth curve models. (DOCX)

**S17 Table. Interaction effects between external childcare by <u>family members</u> and risk on social behavior by age and informant.** Unstandardized coefficients from growth curve models. (DOCX)

**S18 Table. Interaction effects between external childcare by <u>acquaintances</u> and risk on social behavior by age and informant.** Unstandardized coefficients from growth curve models. (DOCX)

**S19 Table. Interaction effects between external childcare by <u>daycare mothers</u> and risk on social behavior by age and informant.** Unstandardized coefficients from growth curve models. (DOCX)

**S20 Table. Interaction effects between external childcare in a <u>daycare center</u> and risk on social behavior by age and informant.** Unstandardized coefficients from growth curve models. (DOCX)

**S21 Table. Interaction effects between external childcare in a <u>playgroup</u> and risk on social behavior by age and informant.** Unstandardized coefficients from growth curve models. (DOCX)

**S22 Table. Interaction effects between external childcare and risk on delinquency and substance use.** Unstandardized coefficients from growth curve models. (DOCX)

**S23 Table. Interaction effects between external childcare and risk on the official delinquency data.** Unstandardized coefficients from regression models. (DOCX)

**S1 Appendix.** (DOCX)

**S2 Appendix.** (DOCX)

**S3 Appendix.** (DOCX)

## Author Contributions

**Conceptualization:** Margit Averdijk, Denis Ribeaud, Manuel Eisner.

**Data curation:** Denis Ribeaud, Manuel Eisner.

**Formal analysis:** Margit Averdijk.

**Funding acquisition:** Margit Averdijk, Denis Ribeaud, Manuel Eisner.

**Investigation:** Denis Ribeaud, Manuel Eisner.

**Methodology:** Margit Averdijk, Denis Ribeaud, Manuel Eisner.

**Project administration:** Margit Averdijk.

**Resources:** Denis Ribeaud, Manuel Eisner.

**Software:** Margit Averdijk.

**Supervision:** Margit Averdijk, Denis Ribeaud, Manuel Eisner.

**Validation:** Margit Averdijk.

**Visualization:** Margit Averdijk.

**Writing – original draft:** Margit Averdijk.

**Writing – review & editing:** Margit Averdijk, Denis Ribeaud, Manuel Eisner.

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
