## [Decision Letter · Decision Letter 0]

13 Aug 2021

PONE-D-21-10331

External childcare and socio-behavioral development in Switzerland: Long-term relations from childhood into young adulthood

PLOS ONE

Dear Dr. Averdijk,

Thank you for submitting your manuscript to PLOS ONE. After careful consideration, we feel that it has merit but does not fully meet PLOS ONE’s publication criteria as it currently stands. Therefore, we invite you to submit a revised version of the manuscript that addresses the points raised during the review process.

As you will see, the reviewers provide thoughtful, but critical feedback on your work. Both reviewers note significant concerns about the inferences to be drawn from this study design. The reviewers are clear on the issues in how the results are interpreted in light of the non-randomization of children into or not into day care settings. Although I do not think that there solutions that will fully address this issue, I encourage you to think deeply about how to clarify the interpretations, such that causal language is eliminated. The reviewers also note multiple concerns with selection and potential non-representativeness of the sample. Some analytic details are in the supplementary material; it may be advisable to include those analyses in the body of the manuscript. While I agree that there may be challenges in using these data to make causal claims, I see these data as being part of the context of understanding this broad area of research, despite the limitations in non-randomized designs. My hope is that the inferences can be tempered and make statements about this critical issue in the field. 

Although I highlight only a small number of the reviewer's comments, please address all comments in full. The reviewers have an appropriately high bar for the scientific evaluation of the work given the developmental and public policy implications of this work. 

We look forward to receiving your revised manuscript.

Kind regards,

Thomas M. Olino

Academic Editor

PLOS ONE

1. Please ensure that your manuscript meets PLOS ONE's style requirements, including those for file naming. The PLOS ONE style templates can be found at https://journals.plos.org/plosone/s/file?id=wjVg/PLOSOne_formatting_sample_main_body.pdf and https://journals.plos.org/plosone/s/file?id=ba62/PLOSOne_formatting_sample_title_authors_affiliations.pdf.

2. Thank you for including your ethics statement: Ethical approval was obtained by the Ethics Committee of the Faculty of Arts and Social Sciences of the University of Zurich

a) Please provide additional details regarding participant consent. In the ethics statement in the Methods and online submission information, please ensure that you have specified (1) whether consent was informed and (2) what type you obtained (for instance, written or verbal, and if verbal, how it was documented and witnessed). If your study included minors, state whether you obtained consent from parents or guardians. If the need for consent was waived by the ethics committee, please include this information.

Additional Editor Comments (if provided):

As you will see, the reviewers provide thoughtful, but critical feedback on your work. Both reviewers note significant concerns about the inferences to be drawn from this study design. The reviewers are clear on the issues in how the results are interpreted in light of the non-randomization of children into or not into day care settings. Although I do not think that there solutions that will fully address this issue, I encourage you to think deeply about how to clarify the interpretations, such that causal language is eliminated. The reviewers also note multiple concerns with selection and potential non-representativeness of the sample. Some analytic details are in the supplementary material; it may be advisable to include those analyses in the body of the manuscript. While I agree that there may be challenges in using these data to make causal claims, I see these data as being part of the context of understanding this broad area of research, despite the limitations in non-randomized designs. My hope is that the inferences can be tempered and make statements about this critical issue in the field.

Although I highlight only a small number of the reviewer's comments, please address all comments in full. The reviewers have an appropriately high bar for the scientific evaluation of the work given the developmental and public policy implications of this work.

Reviewers' comments:

Reviewer's Responses to Questions

**Comments to the Author**

1. Is the manuscript technically sound, and do the data support the conclusions?

Reviewer #1: Partly

Reviewer #2: No

2. Has the statistical analysis been performed appropriately and rigorously? 

Reviewer #1: Yes

Reviewer #2: Yes

3. Have the authors made all data underlying the findings in their manuscript fully available?

Reviewer #1: Yes

Reviewer #2: No

4. Is the manuscript presented in an intelligible fashion and written in standard English?

Reviewer #1: Yes

Reviewer #2: Yes

5. Review Comments to the Author

Reviewer #1: The manuscript "External childcare and socio-behavioral development in Switzerland: Long-term relations from childhood into young adulthood" seeks to answer whether early external childcare -understood as care outside the child’s home- relates to different behavioral outcomes (e.g., externalizing behavior, internalizing problems, prosocial behavior) and risk-taking activities (e.g., delinquency, and substance use) across the lifespan (i.e., from school entry to early adulthood). Although there are strengths to the study, including the use of a large multi-ethnic sample from Switzerland that includes data of life course events and outcomes, as well as the use of multi-informant reported outcomes and the study of different types of childcare arrangements, there are significant limitations that compromise the validity of the inferences made in the study.

Major comments:

Introduction and Results

The introduction section could benefit from a more comprehensive review of the most recent literature on the topic. Given that this article seeks to address a research question that (1) has been very controversial in the field of human development (Dearing and Zachrisson, 2017), and (2) has been most recently approached through quasi-experimental approaches to overcome long-discussed methodological issues of endogeneity/selection bias, I suggest the authors should not only focus on presenting correlational studies but should be more exhaustive in presenting recent research that has used quasi-experimental approaches and strategies to address issues of confounding. Some examples of such studies include:

- Jaffee, S. R., Van Hulle, C., & Rodgers, J. L. (2011). Effects of nonmaternal care in the first 3 years on children’s academic skills and behavioral functioning in childhood and early adolescence: A sibling comparison study. Child Development, 82, 1076–1091. doi:10. 1111/j.1467-8624.2011.01611.x

- Crosby, D. A., Dowsett, C. J., Gennetian, L. A., & Huston, A. C. (2010). A tale of two methods: Comparing regression and instrumental variables estimates of the effects of preschool child care type on the subsequent externalizing behavior of children in low-income families. Developmental Psychology, 46, 1030–1048. doi:10.1037/a0020384

- Baker, M., Gruber, J., & Milligan, K. (2019). The Long-Run Impacts of a Universal Child Care Program. American Economic Journal. Economic Policy, 11(3), 1-26. https://doi.org/10.1257/pol.20170603

Moreover, the authors should also include recent studies that have called into question the internal validity of correlational studies on the relation between time spent in childcare and behavior problems. This recent research has found that conflicting findings on the subject are mostly due to model specifications and the degree of rigor in eliminating selection bias in the models. Indeed, when more rigorous statistical designs have been used to rule out selection bias, the relation between time in child care and behavior problems is, in some studies, no longer statistically significant, suggesting issues of confounding in correlational evidence. Please see the following references for a review of studies:

- Dearing, E., & Zachrisson, H. D. (2017). Concern Over Internal, External, and Incidence Validity in Studies of Child-Care Quantity and Externalizing Behavior Problems. Child Development Perspectives, 11(2), 133–138. https://doi.org/10.1111/cdep.12224

This is also the case for studies that have looked at the association between early child care and behavior problems during adolescence. When studies have used approaches that are less rigorous in ruling out selection bias (using only covariate-adjustment), they have found positive associations between time spent in child care and externalizing behavior problems (e.g., Vandell, Burchinal, and Pierce, 2016), whereas studies using some strategy to address selection, for instance reducing imbalance in observed characteristics by using propensity score matching (e.g., Orri et al. 2019) the relation between child care and behavior problems is even negative. Nonetheless, these important problems related to internal validity are not mentioned in the literature review.

Considering that reviews of the literature clearly demonstrate these issues of selection, I suggest authors revise some of their claims, such as “Our findings on externalizing behavior are largely in line with the international studies discussed earlier, the majority of which have found that more time spent in a daycare center is related to increased externalizing problems at earlier ages (lines 509-511)”. It is not true that the majority of the international evidence indicates such a relation, and most importantly, even if this was the case, the important issue is not whether the majority of papers show a positive association, as if these studies were poorly conducted any inference would be invalid. Instead, the focus should be on internally-valid studies that have used approaches that go beyond traditional covariate adjustment to mitigate issues of selection bias. As already mentioned, such studies tend to find no relation at all and even reductions in behavior problems as children spend more time in child care.

Furthermore, I suggest the authors to revise whether some of the references cited in the article might be equivocally used. For instance, McCartney’s et al. (2010) mentioned that “the evidence linking child care hours with externalizing behavior was equivocal in that results varied across model specifications” (McCartney et al., 2010), and for such reason they tested five causal propositions that “if satisfied, would be consistent with the view that child care hours and externalizing behavior are causally linked.”, which they then show are not satisfied. Since one of the initial claims made in the introduction is that “Prior research on the amount of external childcare has generally concluded that more time in external childcare is related to increased problem behavior.”, citing McCartney’s (2010) article, it seems inaccurate.

Regarding the External childcare in Switzerland subsection, I suggest the authors give a more comprehensive review of the ECEC policies/contexts in Switzerland, and findings are also contextualized accordingly. Switzerland had (still has) the most expensive ECEC in the world (about 50% of disposable income, on average, goes into paying for ECEC), and the country has a very family/mother-unfriendly policies (see Chzhen, Gromada, & Rees, 2019, Are the world’s richest countries family friendly? Policy in the OECD and EU). Although the authors put attention to the counterfactual modes of care, they run the risk of overemphasizing results from a very uncommon context if it is not addressed in more detail.

Results and Discussion

I was disappointed that the authors (from the introduction to the discussion) paid little attention to questions of causality and selection, which are central to the research questions being addressed. Although the authors mention how this study cannot draw causal relations because it uses observational data (lines 325,326), I suggest they clarify that it is not only because of the type of data, but also because the statistical design used does not allow the authors to credibly discard the potential confounders. Even if authors had been capable to rule out reverse causality successfully and prove that parents did not enroll their children into external care due to their problem behavior, there are many other threats to validity that this study does not consider for relying exclusively on covariate-adjustment. Given the implications of a study like this could potentially have, it would not be responsible to present evidence suggesting that child care might be detrimental for young children while relying on a weak empirical strategy that is prone to bias.

Given this, my biggest concern with the paper is the lack of rigor in the language used, as these correlational results could have great implications in practice (i.e., policies on early child care provision, female participation in the labor market). Although the authors acknowledge none of the findings are causal, through the results and discussion they seem to suggest that there might be a causal link. For instance: “Children from less vulnerable backgrounds, however, did not profit from more time in a daycare center or even fared worse” (lines 468-469), or “Though there were some results suggesting beneficial relations (e.g., the association between external family care and prosocial behavior, and decreased externalizing behavior among vulnerable children in daycare), these were offset by less positive ones (e.g., the association between daycare and externalizing, and increased internalizing problems and substance use among vulnerable children in daycare). I encourage authors to address results with much caveat and some careful explanation.

To conclude, while the manuscript could benefit from expanding the literature review and tempering the causal language, the contribution of another correlational study to the field would be minor, considering the long-lasting discussions and evidence showing bias in such studies.

Minor comments:

Methods and Statistical Analyses

Given that ‘time in external childcare’ is the predictor of interest, I suggest moving the information about the data collection using the Event History Calendar back to the article to provide more detailed description of (i) who reported the data, and (ii) the reliability of the measure. Otherwise it becomes less clear who actually reported the childcare information (whether it was the child or the parents) and how the EHC design provides accurate enough data of time spent in childcare that is collected retrospectively. Also, I suggest authors not only give evidence supporting the use of an EHC but also discuss how retrospective self-reported data can bias the results.

It is not clear to me whether the final sample consisted of the 94% of 1,675 first graders that had participated in at least one of the data collections (line 194) or the 1,225 children whose parents had completed the EHC (line 239). I also suggest including the sample size in each of the regression models to be able to compare and check the sample sizes of each model (or clarify that they are the same given that you have done multiple imputation).

I suggest authors discuss further why some of the results showed different directions in the association at different ages: “For example, daycare center care was associated with more self-reported aggression at ages 7 and 8, but the relation flipped at ages 13 and 15, where center care was related to less aggression.”

There are some references that are cited in the text but are not included in the reference list. For example: Vandell et al., 2010 (line 123), Wustmann Seiler et al., 2017 (line 174).

Reviewer #2: The article is well built and well written from a theoretical point of view, and provide a lot of estimations and robustness checks. The article is very (too much?) ambitious, treating several outcomes at different ages, with quite limited, non-representative sample. After reading the summary and the article, I have not a take-away message since results go in several directions according to the informant and child’s age. The major weakness is that the causality between external childcare when the child was young and child’s future development or trend in child’s future development (that the model tests) are not established. How to be sure that the relation observed is causal and not a spurious relationship due to unobserved confounding variable? Even the valuable efforts of authors to control for many of them, this is not enough and I do not belief that growth curve models are particularly adequate for the topic. I would recommend the authors to use policy evaluation methods to treat better causality and to limit the ambitions of the article by limiting the number of outcomes and time periods.

METHOD

I wonder whether the use of growth curve model is adequate for the topic since the event we are interested in (external childcare when child is very young) occur many years before the large period of observation of outcomes (from age 7 to 20). It might affect rather the initial level rather the hereafter trend. I do not see any rationale why the daycare would affect the rate of change at which child developed after the initial level.

If authors keep this method, a short explanation for non-initiated reader is anyway really needed. The authors comment the different parameters without reminding what they mean (inter-individual variability or intra-individual patterns of change over time), so the results are not easily readable for non expert.

The sample size is only mentioned once, and should be precise in each table. Do the sample size available is the same for all regressions? If not, what has been the choice to keep the common individuals interviewed each wave or to keep maximum individuals at each wave?

The age range from 7 to 20 is very large

RESULTS

I would like to see a description of selection into external childcare. Who are the parents who use external childcare and which type of childcare?

Some results are weird, particularly when they differ by informant. They seem sensitive to measurement, so I have doubts about their validity.

A discussion is missing about the non representative sample. As it is a sample based on a town, it may be possible to control for the daycare attended more precisely.

The fact that more internalizing problems and substance problems are only observed for children attending daycare coming from vulnerable background put doubts on the role of unobserved characteristics. Who are these children? Do they attend same daycare than others? Do they have the same life-course?

More importantly, several unobserved characteristics are possible and strong mediator of results: for instance whether the mother was working part-time full-time (probably by financial necessity for children coming from disadvantaged background) part-time or not working at all, the type of kindergarten attended (of worse quality), the instability of family structure (dissolution and the arrival of a step-parent) from Kindergarden or hereafter, etc.

Minor remarks:

Why is the rationale of precising that the sample is multi-ethnic in the summary since no much emphasis on this point is put in the article?

The limitations section should come back on data, and selection in use of external child care.

6. PLOS authors have the option to publish the peer review history of their article (what does this mean?). If published, this will include your full peer review and any attached files.

Reviewer #1: No

Reviewer #2: No

---

## [Author Response · Author response to Decision Letter 0]

26 Sep 2021

Please refer to the uploaded Cover Letter and Response to Reviewers document

---

## [Decision Letter · Decision Letter 1]

24 Nov 2021

PONE-D-21-10331R1External childcare and socio-behavioral development in Switzerland: Long-term relations from childhood into young adulthoodPLOS ONE

Dear Dr. Averdijk,

Thank you for submitting your manuscript to PLOS ONE. After careful consideration, we feel that it has merit but does not fully meet PLOS ONE’s publication criteria as it currently stands. Therefore, we invite you to submit a revised version of the manuscript that addresses the points raised during the review process.

My apologies for my error in the previous letter. 

I was able to obtain the review from one of the original reviewers, but the other was unable to provide comments. In my reading of the manuscript, I see much greater clarity in the ways that you highlight what your study can and cannot conclude based on the design employed in the study. At the same time, the Reviewer is asking for transparency in the methods used by previous studies, particularly those studies that used methods permitting drawing causal inferences. If there were observational studies that appropriately drew causal inferences, some additional clarifications about why those methods could not be used in your work would be helpful. 

We look forward to receiving your revised manuscript.

Kind regards,

Thomas M. Olino

Academic Editor

PLOS ONE

Journal Requirements:

Reviewers' comments:

Reviewer's Responses to Questions

**Comments to the Author**

1. If the authors have adequately addressed your comments raised in a previous round of review and you feel that this manuscript is now acceptable for publication, you may indicate that here to bypass the “Comments to the Author” section, enter your conflict of interest statement in the “Confidential to Editor” section, and submit your "Accept" recommendation.

Reviewer #1: (No Response)

2. Is the manuscript technically sound, and do the data support the conclusions?

Reviewer #1: No

3. Has the statistical analysis been performed appropriately and rigorously? 

Reviewer #1: Yes

4. Have the authors made all data underlying the findings in their manuscript fully available?

Reviewer #1: Yes

5. Is the manuscript presented in an intelligible fashion and written in standard English?

Reviewer #1: Yes

6. Review Comments to the Author

Reviewer #1: 1. Although the authors have included the suggested literature, I’m worried that the conclusion inferred from the causal evidence and from the most recent reviews (e.g., Dearing & Zachrisson, 2017; Huston, 2015; McCartney et al., 2010) is still wanting. It is not that “different methods may yield to different results” but that more rigorous methods with better identification strategies to address selection bias lead to more internally valid results. I am afraid this idea is still not well addressed in the article and that the reader might potentially draw the wrong conclusions from the article because of this.

This is the case in statements such as: “Past literature reviews on the link between the quantity of external childcare and child development have concluded that external childcare is related to increased problem behavior (see Dearing & Zachrisson, 2017, for an overview).”, which does not clarify that this is the case when less rigorous methodologies such as covariate-adjustment/OLS are implemented but that the opposite relation is found when selection bias is addressed through quasi-experimental approaches. Another example is: “Other studies have used other methods, which have their own unique advantages and disadvantages, such as sibling comparisons, propensity score matching, and instrumental variables and shown that different methods may yield different results (e.g., McCartney et al., 2010).”

2. I strongly suggest that the authors relate the specific methods used in the articles they mentioned to make particularly strong claims. Again, this kinds of statements that have strong policy implications must be stated with enough context. For instance, "the previously mentioned NICHD study found that even though low-quality external care is more often associated with problem behavior than high-quality care, external care was related to more externalizing behavior at all levels of quality (NICHD Early Child Care Research Network, 2003)." or "A Swiss study showed that an increased availability of external childcare provisions would affect parental labor participation only marginally (Stern et al., 2018)." What analytical strategy was this study using? What was the identification strategy used? What was the sample studied? Was it nationally-representative?

3. Although the data used in this study is rich in that it follows children until adolescence and early adulthood and it includes multiple outcome variables, I want to reiterate that this study fails to address selection effects, making it difficult to draw meaningful conclusions. While the authors employ covariates and some initial robustness checks, controlling for observed characteristics, the growth curve model employed is insufficient to establish internally valid estimates given that potential unobserved characteristics that relate to both time spent in external child care and the outcomes might still not be accounted for. It is exactly for this reason that recent research has moved away from correlational designs to address a topic that is very sensitive in terms of policy implications for early childhood care and education supply.

7. PLOS authors have the option to publish the peer review history of their article (what does this mean?). If published, this will include your full peer review and any attached files.

Reviewer #1: No

---

## [Author Response · Author response to Decision Letter 1]

17 Jan 2022

All of our responses to the reviewer's and editor's comments can be found in the Cover letter and Response to Reviewers document.

---

## [Editor Report · Decision Letter 2]

24 Jan 2022

External childcare and socio-behavioral development in Switzerland: Long-term relations from childhood into young adulthood

PONE-D-21-10331R2

Dear Dr. Averdijk,

We’re pleased to inform you that your manuscript has been judged scientifically suitable for publication and will be formally accepted for publication once it meets all outstanding technical requirements.

Kind regards,

Thomas M. Olino

Academic Editor

PLOS ONE
---

## [Editor Report · Acceptance letter]

16 Feb 2022

PONE-D-21-10331R2 

External childcare and socio-behavioral development in Switzerland: Long-term relations from childhood into young adulthood 

Dear Dr. Averdijk:

I'm pleased to inform you that your manuscript has been deemed suitable for publication in PLOS ONE. Congratulations! Your manuscript is now with our production department. 

Kind regards, 

on behalf of

Dr. Thomas M. Olino 

Academic Editor

PLOS ONE